# Intraday reliability, sensitivity, and minimum detectable change of National Physical Fitness Measurement for preschool children in China

**Hua Fang[1], Indy Man Kit Ho**[2,3]*

**1** School of Strength and Conditioning Training, Beijing Sports University, Beijing, China, **2** Department of Sports and Recreation, Technological and Higher Education Institute of Hong Kong (THEi), Hong Kong, **3** The Asian Academy for Sports and Fitness Professionals

* indymankit@hotmail.com

**Data Availability Statement:** Data are available within the Supporting Information files and at Figshare (https://doi.org/10.6084/m9.figshare.13191491.v1).

## Abstract

China General Administration of Sport has published and adopted the National Physical Fitness Measurement (NPFM—preschool children version) since 2000. However, studies on intraday reliability, sensitivity, and minimum detectable change (MDC) are lacking. This study aimed to investigate and compare the reliability, sensitivity, and MDC values of NPFM in preschool children between the ages of 3.5 to 6 years. Six items of NPFM including 10-m shuttle run, standing long jump, balance beam walking, sit-and-reach, tennis throwing, and double-leg timed hop, were tested for 209 Chinese kindergarten children in Beijing in the morning. Intraday relative reliability was tested using intraclass correlation coefficient ($ICC_{3,1}$) with a 95% confidence interval while absolute reliability was expressed in standard error of measurement (SEM) and percentage of coefficient of variation (CV%). Test sensitivity was assessed by comparing the smallest worthwhile change (SWC) with SEM, while MDC values with 95% confidence interval ($MDC_{95}$) were established. Measurements in most groups, except 10-m shuttle run test ($ICC_{3,1}$: 0.56 to 0.74 [moderate]) in the 3.5 to 5.5-year-old groups, balance beam test in 4- and 5-year-old ($ICC_{3,1}$: 0.33 to 0.35 [poor]) and 5.5-year-old ($ICC_{3,1} = 0.68$ [moderate]) groups, and double-leg timed hop test ($ICC_{3,1} = 0.67$ [moderate]) in the 4.5-year-old group, demonstrated good to excellent relative reliability ($ICC_{3,1}$: 0.77 to 0.97). The balance beam walking test showed poor absolute reliability in all the groups (SEM%: 11.76 to 22.28 and CV%: 15.40 to 24.78). Both standing long jump and sit-and-reach tests demonstrated good sensitivity (SWC > SEM) in all subjects group, boys, and girls. Pairwise comparison revealed systematic bias with significantly better performance in the second trial ($p<0.01$) of all the tests with moderate to large effect size.

## Introduction

Evaluation of physical fitness level is vital for recognizing health conditions and predicting the risk of chronic diseases for populations [1–3]. Therefore, many countries have developed and adopted a battery of national fitness tests with health-related fitness components, such as muscular strength, flexibility, cardiorespiratory endurance, and body composition [4–6]. Similarly,

**Funding:** The authors received no specific funding for this work.

**Competing interests:** The authors have declared that no competing interests exist.

for preschool children, a battery of or protocol for comprehensive physical fitness tests is essential to monitor trends and severity of obesity issues and determine adequacy of physical activities [7]. Therefore, the China General Administration of Sport has published and adopted the National Physical Fitness Measurement (NPFM) since 2000 while its preschool children version was developed concurrently with six assessment items, namely, 10-m shuttle run (SRT), standing long jump (SLJ), balance beam walking, sit-and-reach, tennis throwing (TT), and double-leg timed hop (DTH) tests [8].

NPFM is a longitudinal study promoted by the Chinese government to observe health and fitness conditions from large samples of populations. The test results can be used to compare findings of preschoolers of similar ages from other countries. Similarly, the government uses the test results of children to understand the variation of physical fitness competence among cities, evaluate outcomes and performance of the "national fitness program" being promoted to Chinese citizens, and provide scientific evidence for updating such program with justifications and rationales. Apart from determining the fitness level, NPFM for preschoolers can also be used to identify motor performance, screen underdeveloped children for further evaluation, and enhance exercise motivation. In this regard, a battery of tests adopting reliable and useful testing items is crucial to provide meaningful results for further analyses. Therefore, reliable and valid measurements with sufficient sensitivity are vital.

Previous studies showed excellent reliability of FITness testing in PREschool children (PREFIT) in Spain using Bland-Altman method, intra-class correlation coefficient (ICC) and the comparison of mean differences [6, 9]. Meanwhile, the systematic review from Ortega et al. [4] reported that 4 x 10 m shuttle-run test has provides reliable measures in speed and agility related fitness for preschoolers aged 4 to 5 years (ICC: 0.52 to 0.92) and one-leg-stance test is a popular and reliable test for assessing the balance of preschool children (ICC: 0.73 to 0.99). In addition, the standing long jump test used in testing 4- and 5-year-old preschool children showed acceptable relative reliability (ICC: 0.65 to 0.89). Regarding the studies using Chinese NPFM, the level of physical fitness and activity of preschool children in Shanghai was reported recently [7, 10]. However, investigations on the reliability, sensitivity, and minimum detectable change (MDC) values of testing items in NPFM are lacking. As preschool children undergo rapid development in motor skills and physical fitness [11, 12], Latorre Román et al. [5] demonstrated the remarkable variation in the physical fitness of preschool children of different ages and large variance of performance within groups. Therefore, the reliability and sensitivity of the test battery in NPFM that assess Chinese preschool children of different ages are speculated to be varied also with such immature motor development and unstable motor performance. This study aimed to investigate and compare the reliability, sensitivity, and MDC values of NPFM in preschool children between the ages of 3.5 and 6 years to solve these problems.

## Materials and methods

### Subjects

This study was approved by the institutional review board of Beijing Sports University and conducted according to the Declaration of Helsinki by strictly following the protocol of NPFM (preschool children version), which was published by the government of China [8]. Two hundred and nine Chinese kindergarten children (111 boys and 98 girls) were recruited on a voluntary basis. Anthropometric data, such as age, body height, and body mass, of different genders and age groups are listed in Table 1. Subjects were further divided into the following subgroups according to their chronological ages: $\leq 3.5$ (n = 31) $< 4$, $\leq 4$ (n = 22) $< 4.5$, $\leq 4.5$ (n = 43) $< 5$, $\leq 5$ (n = 24) $< 5.5$, $\leq 5.5$ (n = 45) $< 6$, and $\leq 6$ (n = 44) years old. The

**Table 1. Anthropometric data of different genders and age groups.**

| Group | Age (year) | Height (cm) | Weight (kg) |
|---|---|---|---|
| | mean±SD | mean±SD | mean±SD |
| All subjects (n = 209) | 5.14±0.88 | 112.61±7.63 | 20.53±4.18 |
| Boys (n = 111) | 5.13±0.91 | 113.09±7.99 | 20.81±4.45 |
| Girls (n = 98) | 5.16±0.86 | 112.06±7.21 | 20.21±3.84 |
| 3.5-year-old (n = 31) | 3.76±0.12 | 102.30±3.19 | 16.47±1.32 |
| 4-year-old (n = 22) | 4.22±0.12 | 106.17±4.04 | 18.66±3.66 |
| 4.5-year-old (n = 43) | 4.76±0.16 | 110.03±4.19 | 19.25±2.76 |
| 5-year-old (n = 24) | 5.24±0.14 | 112.00±4.01 | 20.12±3.39 |
| 5.5-year-old (n = 45) | 5.76±0.14 | 118.51±4.88 | 23.32±4.51 |
| 6-year-old (n = 44) | 6.28±0.19 | 119.89±4.62 | 22.94±3.57 |

classification system was based on principles and instructions of NPFM [8]. Three-year-old preschool children were not included because tests were conducted in the second semester of their academic year. Therefore, the youngest group in this study was composed of 3.5-year-old children while the oldest group comprised students above 6 years old. Informed written consent containing experimental procedures, potential benefits, and explained risks was obtained from each child's parents. Any subject with diagnosed illness or identified deformity that may potentially limit the completion of NPFM was excluded to enhance the testing accuracy and minimize the risk of injuries.

## Procedures

NPFM was conducted by trained research assistants on a synthetic rubber surface at the outdoor playground of a kindergarten school in Beijing in the morning. Subjects performed six mandatory testing items in randomized order. According to the current NPFM guidelines [8], no previous familiarization session was given. After providing verbal instructions and demonstrations, each subject performed two trials for each measurement item with at least one minute of rest in between while all the tests were conducted by the same rater.

## Double-leg timed hop test

Ten rectangular soft blocks (10 cm [length] × 5 cm [width] × 5 cm [height]) were placed in a straight line at 50 cm apart from each other and used as barriers. Prior to the start of DTH, posture and position of subjects were standardized as standing with their feet together at 20 cm behind the first block. Subjects were required to jump over all the barriers as fast as possible after the start signal was given. The time to complete jumping over all the blocks was recorded while any trial with foot stepping or kicking on the barrier was regarded as fail. Subjects had to redo the test for failed trials. The test results were measured in seconds [8].

## Standing long jump test

Subjects stood behind the starting line as the ready position and were instructed to jump as far as possible with arms swinging and landing with both feet for the SLJ test. The distance was recorded in centimeters using a tape measure from the starting line to the heel of the rear landing foot [8].

## Tennis throwing test

Subjects stood behind the starting line and threw a tennis ball forward as far as possible for the TT test. Any trial with the foot stepping on or over the starting line during or after throwing was regarded as fail and redoing the failed attempt was required. The testing results were measured from the starting line to the first landing point of the ball in meters [8].

## 10-m shuttle run test

An object with similar height to the majority of subjects was set at a distance of 10 m from the starting line to ensure minimum change of running posture. Each subject was instructed to reach out an arm and touch the object before turning. Subjects were required to run at full speed after the "action" signal was given, touch the target object, and run back to the starting line as fast as possible, with the results recorded in seconds [8].

## Balance beam walking test

Subjects were required to walk along a 3 m-long, 10 cm-wide, and 30 cm-high balance beam as fast as possible with arms kept at a 90˚ abduction position. The completion time was recorded in seconds. In the event that a subject falls down from the beam during the walking process, the test was regarded as fail and a make-up trial was needed [8].

## Sit-and-reach test

Subjects sat on the ground with bare feet together and knees straight. Before starting, the soles of their feet should press against the edge of the sit-and-reach box and such contact position was regarded as zero point. Subjects were required to bend their trunks forward and push the moveable marker of scale plate with their fingertips as far as possible without bending their knees. The distance from the start point to the place where the marker stopped was recorded in centimeters. Trials with a stopped marker that failed to pass the zero point were recorded as negative values [8].

## Statistical analyses

The results were presented as mean and standard deviation (SD) while the intraday relative reliability was tested using intraclass correlation coefficient with two-way mixed-effects model and single measurement ($ICC_{3,1}$) with a 95% confidence interval (95% CI) using SPSS 24.0 for Windows (SPSS Inc.; Chicago, IL). ICC values of less than 0.5, between 0.5 and 0.75, between 0.75 and 0.9, and larger than 0.90 are regarded as poor, moderate, good, and excellent relative reliability, respectively [13]. Meanwhile, the standard error of measurement (SEM) or typical error according to Hopkins (2000) [14] and MDC with 95% CI ($MDC_{95}$) were obtained using the following formulas: $SEM = SD/\sqrt{2}$ and $MDC_{95} = SEM \times \sqrt{2} \times 1.96$, where SD used for calculating SEM is the standard deviation of the difference between trials [15]. SEM% is the percentage of mean cumulative test–retest scores [16]. Coefficient of variation expressed in the percentage of the mean score of individuals (CV%) together with SEM% was calculated to indicate the absolute reliability [17], and SEM% and CV% below 10% were deemed acceptable [16, 17]. Smallest worthwhile change (SWC) is calculated using $0.2 \times SD$, where SD represents the between-subject standard deviation of the best trial, to verify the usefulness of each test further. Test sensitivity was assessed by comparing the SWC and SEM, where SEM below SWC indicates "good" sensitivity, SEM similar to SWC is rated "satisfactory," and SEM higher than SWC is deemed to have "marginal" sensitivity [18, 19].

Paired sample t-tests were used to determine the significant difference between trials and confirm the existence of systematic bias. Effect size (Cohen's *d*) further provided the magnitude of difference while significance level for all statistical tests was set to $p<0.05$ and heteroscedasticity was determined.

## Results

### Reliability and sensitivity analysis for all subjects, boys, and girls

Heteroscedasticity was nonsignificant in all the groups (*p*: 0.11 to 0.98). Table 2 shows good to excellent ICCs (0.77 to 0.97) of all the measurements in the groups of all subjects, boys, and girls. However, the balance beam walking test demonstrated poor absolute reliability for the groups of all ages (SEM% = 18.05 and CV% = 20.43), boys (SEM% = 17.96 and CV% = 20.47), and girls (SEM% = 18.10% and CV% = 20.38%). $MDC_{95}$ values in the balance beam walking test for groups of all subjects, boys, and girls showed a minimum threshold of 4.09, 3.99, and 4.18 s, respectively, which are beyond the random measurement error with a 95% confidence level.

SLJ demonstrated good sensitivity in the group of all subjects (SWC = 4.54 > SEM = 3.81), boys (SWC = 4.68 > SEM = 3.94), and girls (SWC = 4.33 > SEM = 3.67). Similarly, the sit-and-reach test showed good sensitivity in the group of all subjects (SWC = 0.90 > SEM = 0.63), boys (SWC = 0.77 > SEM = 0.68), and girls (SWC = 0.89 > SEM = 0.41). Only the boys group (SWC = 0.40 > SEM = 0.30) exhibited good sensitivity in the TT test, while satisfactory sensitivity was observed in all the subjects (SWC = 38 ≈ SEM = 0.36).

**Table 2. ICC, CV%, SEM, SWC, and $MDC_{95}$ and classification of sensitivity of all subjects, boys, and girls.**

| Group | Testing Items | Mean±SD | | ICC (95% CI) | CV% | SEM (%) | SWC | $MDC_{95}$ | Sensitivity |
|---|---|---|---|---|---|---|---|---|---|
| | | Trial 1 | Trial 2 | | | | | | |
| All subjects (n = 209) | 10-m SRT (s) | 8.29±1.38 | 7.77±1.26 | 0.84 (0.44–0.93) | 6.27 | 0.42 (5.19) | 0.25 | 1.16 | Marginal |
| | SLJ (cm) | 89.26±22.82 | 93.11±22.85 | 0.96 (0.87–0.98) | 5.04 | 3.81 (4.18) | 4.54 | 10.56 | Good |
| | TT (m) | 4.64±1.93 | 5.07±1.89 | 0.94 (0.75–0.98) | 10.51 | 0.36 (7.47) | 0.38 | 1.01 | Satisfactory |
| | DTH (s) | 7.32±2.38 | 6.62±2.15 | 0.90 (0.56–0.96) | 8.79 | 0.53 (7.63) | 0.42 | 1.47 | Marginal |
| | Sit-and-reach (cm) | 8.04±4.47 | 9.43±4.50 | 0.94 (0.19–0.98) | 18.26 | 0.63 (7.17) | 0.90 | 1.74 | Good |
| | Balance beam walking (s) | 8.88±4.39 | 7.45±4.26 | 0.84 (0.60–0.92) | 20.43 | 1.47 (18.05) | 0.80 | 4.09 | Marginal |
| Boys (n = 111) | 10-m SRT (s) | 8.14±1.18 | 7.64±1.16 | 0.80 (0.40–0.91) | 6.40 | 0.42 (5.32) | 0.22 | 1.16 | Marginal |
| | SLJ (cm) | 91.43±23.53 | 95.38±23.63 | 0.96 (0.87–0.98) | 5.06 | 3.94 (4.22) | 4.68 | 10.93 | Good |
| | TT (m) | 5.19±2.00 | 5.58±1.96 | 0.96 (0.78–0.99) | 7.78 | 0.30 (5.56) | 0.40 | 0.83 | Good |
| | DTH (s) | 7.30±2.43 | 6.67±2.24 | 0.92 (0.63–0.97) | 8.20 | 0.50 (7.13) | 0.44 | 1.38 | Marginal |
| | Sit-and-reach (cm) | 6.01±3.56 | 7.75±3.84 | 0.87 (−0.01–0.97) | 28.32 | 0.68 (9.89) | 0.77 | 1.89 | Good |
| | Balance beam walking (s) | 8.82±3.73 | 7.23±3.54 | 0.77 (0.40–0.89) | 20.47 | 1.44 (17.96) | 0.65 | 3.99 | Marginal |
| Girls (n = 98) | 10-m SRT (s) | 8.46±1.57 | 7.93±1.36 | 0.86 (0.45–0.95) | 6.11 | 0.41 (5.06) | 0.27 | 1.15 | Marginal |
| | SLJ (cm) | 86.80±21.86 | 90.53±21.76 | 0.96 (0.86–0.98) | 5.02 | 3.67 (4.14) | 4.33 | 10.18 | Good |
| | TT (m) | 4.03±1.64 | 4.51±1.63 | 0.90 (0.63–0.96) | 13.60 | 0.42 (9.90) | 0.33 | 1.71 | Marginal |
| | DTH (s) | 7.33±2.32 | 6.57±2.06 | 0.88 (0.47–0.95) | 9.45 | 0.57 (8.16) | 0.40 | 1.57 | Marginal |
| | Sit-and-reach (cm) | 10.35±4.29 | 11.33±4.46 | 0.97 (0.28–0.99) | 6.86 | 0.41 (3.75) | 0.89 | 1.13 | Good |
| | Balance beam walking (s) | 8.95±5.06 | 7.70±4.96 | 0.88 (0.74–0.94) | 20.38 | 1.51 (18.10) | 0.95 | 4.18 | Marginal |

Abbreviations: SRT, Shuttle Run Test; SLJ, Standing Long Jump; TT, Tennis Throwing; DTH, Double-leg Timed Hop; s, second; cm, centimeter; m, meter; SD, Standard Deviation; ICC, Intraclass Correlation Coefficient; CV%, Percentage of Coefficient of Variation; CI, Confidence Interval; SEM, Standard Error of Measurement; SWC, Smallest Worthwhile Change; $MDC_{95}$, Minimum Detectable Change in 95% CI.

### Reliability and sensitivity analysis for different age groups

Intraday reliability in ICC, CV%, SEM, SWC, and MDC$_{95}$ data and classification of sensitivity in 3.5-, 4-, 4.5-, 5-, 5.5-, and 6-year-old subjects are presented in Table 3. The majority of measurements showed good to excellent relative reliability (ICC: 0.79 to 0.95), except the 10-m SRT (ICC: 0.67 to 0.73 [moderate]) in three groups (3.5-, 4-, and 5-year-old subjects), balance

**Table 3. ICC, CV%, SEM, SWC, and MDC$_{95}$ and classification of sensitivity in 3.5-, 4-, 4.5-, 5-, 5.5-, and 6-year-old subjects.**

| Group | Testing Items | Mean±SD | | ICC (95% CI) | CV% | SEM (%) | SWC | MDC$_{95}$ | Sensitivity |
|---|---|---|---|---|---|---|---|---|---|
| | | Trial 1 | Trial 2 | | | | | | |
| 3.5-year-old | 10-m SRT (s) | 9.85±1.35 | 9.09±1.26 | 0.67 (0.15–0.86) | 7.79 | 0.61 (6.47) | 0.24 | 1.70 | Marginal |
| | SLJ (cm) | 65.26±18.76 | 68.45±18.01 | 0.93 (0.83–0.97) | 7.04 | 4.39 (6.57) | 3.61 | 12.18 | Marginal |
| | TT (m) | 2.79±1.18 | 3.32±1.12 | 0.80 (0.25–0.93) | 17.81 | 0.39 (12.63) | 0.23 | 1.07 | Marginal |
| | DTH (s) | 9.73±2.71 | 8.54±2.43 | 0.85 (0.12–0.96) | 10.73 | 0.66 (7.22) | 0.49 | 1.83 | Marginal |
| | Sit-and-reach (cm) | 8.50±3.35 | 9.74±3.52 | 0.88 (0.39–0.96) | 9.83 | 0.87 (9.59) | 0.70 | 2.42 | Marginal |
| | Balance beam walking (s) | 13.23±5.90 | 11.32±6.46 | 0.83 (0.59–0.92) | 23.87 | 2.30 (18.78) | 1.23 | 6.39 | Marginal |
| 4-year-old | 10-m SRT (s) | 8.93±1.02 | 8.13±0.97 | 0.68 (−0.08–0.91) | 7.18 | 0.33 (3.84) | 0.19 | 0.91 | Marginal |
| | SLJ (cm) | 65.95±18.97 | 71.14±17.83 | 0.91 (0.60–0.97) | 8.06 | 4.43 (6.47) | 3.70 | 12.29 | Marginal |
| | TT (m) | 3.61±1.41 | 4.11±1.45 | 0.91 (0.26–0.98) | 11.90 | 0.28 (7.35) | 0.29 | 0.79 | Satisfactory |
| | DTH (s) | 9.64±2.66 | 8.93±2.58 | 0.88 (0.65–0.95) | 10.05 | 0.80 (8.65) | 0.50 | 2.22 | Marginal |
| | Sit-and-reach (cm) | 6.16±2.41 | 7.25±2.81 | 0.87 (0.15–0.97) | 11.63 | 0.61 (9.09) | 0.56 | 1.69 | Marginal |
| | Balance beam walking (s) | 10.86±2.90 | 9.60±2.78 | 0.33 (−0.06–0.65) | 22.73 | 2.28 (22.28) | 0.46 | 6.32 | Marginal |
| 4.5-year-old | 10-m SRT (s) | 8.87±0.97 | 8.39±0.95 | 0.73 (0.29–0.88) | 5.88 | 0.42 (4.81) | 0.19 | 1.15 | Marginal |
| | SLJ (cm) | 83.16±12.62 | 86.49±12.14 | 0.88 (0.69–0.95) | 4.62 | 3.63 (4.28) | 2.40 | 10.06 | Marginal |
| | TT (m) | 4.58±1.70 | 5.04±1.78 | 0.94 (0.53–0.98) | 9.87 | 0.30 (6.20) | 0.35 | 0.83 | Good |
| | DTH (s) | 6.89±1.31 | 5.96±1.21 | 0.67 (−0.05–0.89) | 11.87 | 0.48 (7.52) | 0.23 | 1.34 | Marginal |
| | Sit-and-reach (cm) | 9.46±4.27 | 10.70±4.32 | 0.95 (0.08–0.99) | 11.06 | 0.43 (4.24) | 0.86 | 1.18 | Good |
| | Balance beam walking (s) | 10.78±4.68 | 8.28±4.75 | 0.83 (−0.02–0.95) | 24.08 | 1.07 (11.25) | 0.94 | 2.97 | Marginal |
| 5-year-old | 10-m SRT (s) | 8.53±0.75 | 8.02±0.53 | 0.56 (−0.02–0.82) | 5.40 | 0.34 (4.16) | 0.11 | 0.95 | Marginal |
| | SLJ (cm) | 84.96±14.60 | 89.71±16.58 | 0.90 (0.53–0.97) | 5.24 | 3.70 (4.24) | 3.19 | 10.26 | Marginal |
| | TT (m) | 3.88±1.31 | 4.28±1.30 | 0.91 (0.50–0.97) | 9.76 | 0.30 (7.31) | 0.26 | 0.83 | Marginal |
| | DTH (s) | 6.95±3.23 | 6.15±2.90 | 0.94 (0.55–0.98) | 11.47 | 0.55 (8.46) | 0.58 | 1.54 | Satisfactory |
| | Sit-and-reach (cm) | 8.72±5.05 | 10.36±5.27 | 0.94 (0.06–0.99) | 18.20 | 0.58 (6.10) | 1.05 | 1.61 | Good |
| | Balance beam walking (s) | 7.16±1.85 | 5.70±1.56 | 0.35 (−0.05–0.66) | 24.78 | 1.24 (19.35) | 0.30 | 3.45 | Marginal |
| 5.5-year-old | 10-m SRT (s) | 7.52±0.84 | 7.12±0.86 | 0.74 (0.34–0.89) | 6.11 | 0.37 (4.99) | 0.17 | 1.01 | Marginal |
| | SLJ (cm) | 103.51±12.92 | 107.0±13.74 | 0.90 (0.70–0.96) | 3.77 | 3.57 (3.39) | 2.68 | 9.89 | Marginal |
| | TT (m) | 5.72±1.78 | 6.07±1.73 | 0.92 (0.81–0.97) | 8.40 | 0.43 (7.22) | 0.35 | 1.18 | Marginal |
| | DTH (s) | 6.25±0.87 | 5.80±0.88 | 0.79 (0.12–0.93) | 6.29 | 0.28 (4.68) | 0.17 | 0.78 | Marginal |
| | Sit-and-reach (cm) | 8.37±4.69 | 9.83±4.52 | 0.94 (0.09–0.99) | 26.05 | 0.56 (6.19) | 0.90 | 1.56 | Good |
| | Balance beam walking (s) | 6.48±1.78 | 5.68±1.59 | 0.68 (0.33–0.84) | 16.06 | 0.84 (13.86) | 0.31 | 2.34 | Marginal |
| 6-year-old | 10-m SRT (s) | 6.97±0.88 | 6.60±0.87 | 0.80 (0.37–0.92) | 5.74 | 0.31 (4.59) | 0.17 | 0.86 | Marginal |
| | SLJ (cm) | 111.55±14.03 | 115.57±14.34 | 0.90 (0.64–0.96) | 3.70 | 3.59 (3.17) | 2.83 | 9.96 | Marginal |
| | TT (m) | 5.85±1.72 | 6.23±1.67 | 0.92 (0.76–0.97) | 7.87 | 0.40 (6.70) | 0.33 | 1.12 | Marginal |
| | DTH (s) | 6.16±0.99 | 5.87±1.05 | 0.91 (0.56–0.97) | 4.87 | 0.23 (3.77) | 0.21 | 0.63 | Satisfactory |
| | Sit-and-reach (cm) | 6.57±5.05 | 8.14±5.01 | 0.94 (0.13–0.99) | 26.60 | 0.65 (8.82) | 1.00 | 1.80 | Good |
| | Balance beam walking (s) | 6.38±2.27 | 5.59±2.56 | 0.87 (0.57–0.95) | 15.40 | 0.70 (11.76) | 0.47 | 1.95 | Marginal |

Abbreviations: SRT, Shuttle Run Test; SLJ, Standing Long Jump; TT, Tennis Throwing; DTH, Double-leg Timed Hop; SD, Standard Deviation; ICC, Intraclass Correlation Coefficient; CI, Confidence Interval; CV%, Percentage of Coefficient of Variation; SEM, Standard Error of Measurement; SWC, Smallest Worthwhile Change; MDC$_{95}$, Minimum Detectable Change in 95% CI.

beam test (ICC: 0.33 to 0.68 [poor to moderate]) in 4-, 5-, and 5.5-year-old subjects, and DTH (ICC = 0.67 [moderate]) in 4.5-year-old subjects. However, according to SEM% and CV% values, the balance beam walking test demonstrated poor absolute reliability (SEM%: 11.25 to 22.28 and CV%: 15.40 to 24.78) for all the age groups.

The comparison of SWC and SEM values showed that most measurements demonstrated only marginal sensitivity, except the TT test of 4.5-year-old subjects (SWC = 0.35 > SEM = 0.30) and the sit-and-reach test of 4.5- (SWC = 0.86 > SEM = 0.43), 5- (SWC = 1.05 > SEM = 0.58), 5.5- (SWC = 0.90 > SEM = 0.56), and 6-year-old (SWC = 1.00 > SEM = 0.65) subjects. Meanwhile, satisfactory sensitivity was observed in the TT test of 4-year-old subjects (SWC = 0.29 ≈ SEM = 0.28) and DTH in 5- (SWC = 0.58 ≈ SEM = 0.55) and 6-year-old (SWC = 0.21 ≈ SEM = 0.23) subjects.

### Differences and effect size between trials of all measurements

The results of pairwise sample t-test (Table 4) showed a significant difference between trials for all the measurements of the 10-m SRT ($p < 0.01$ and $d = 0.87$ [large]), SLJ ($p < 0.01$ and $d = 0.71$ [moderate]), TT ($p < 0.01$ and $d = 0.84$ [large]), DTH ($p < 0.01$ and d = 0.92 [large]), sit-and-reach ($p < 0.01$ and $d = 1.57$ [large]), and balance beam walking ($p < 0.01$ and $d = 0.69$ [moderate]) tests.

## Discussion

This study primarily aimed to set up the intraday reliability, MDC, and sensitivity of six key testing items of NPFM by comparing between trials. The systematic bias of observed differences, such as potential of the learning effect to lead to a higher degree of familiarity of the selected measurement, insufficient recovery from the previous trial that induces the fatigue effect to subsequent attempts, and different emotional statuses or motivation levels, can be detected when intertrial reliability is determined [17].

The findings shown in Table 2 indicated that all the testing items generally demonstrate a good to excellent relative reliability in preschool children. ICC is commonly used to assess the reliability of a measurement or testing method, wherein values over 0.90 are regarded as excellent relative test–retest reliability. Tests with excellent ICCs exhibit good stability and consistency of measurement over time and low measurement error [20]. However, previous studies reported limitations, such as inter subject variability that can potentially affect the result and overestimated ICC values in a typically heterogeneous population, in the use of ICC alone [21]. Therefore, measurements with excellent relative reliability do not necessarily ensure consistent intertrial performance. Calculations of SEM and CV% were further recommended to obtain within-subject variation in addition to measuring ICCs and confirm the absolute

**Table 4. Differences in mean values between trials of measurements.**

| Group | Testing Items | Difference in Trials 2 and 1 (SD) | p | Effect Size Cohen's *d* | |
|---|---|---|---|---|---|
| All subjects | 10-m SRT (s) | −0.52 (0.60) | < 0.01 | −0.87 | Large |
| | SLJ (cm) | 3.85 (5.39) | < 0.01 | 0.71 | Moderate |
| | TT (m) | 0.43 (0.51) | < 0.01 | 0.84 | Large |
| | DTH (s) | −0.70 (0.75) | < 0.01 | −0.92 | Large |
| | Sit-and-reach (cm) | 1.39 (0.89) | < 0.01 | 1.57 | Large |
| | Balance beam (s) | −1.43 (2.08) | < 0.01 | −0.69 | Moderate |

Abbreviations: SRT, Shuttle Run Test; SLJ, Standing Long Jump; TT, Tennis Throwing; DTH, Double-leg Timed Hop; SD, Standard Deviation

reliability [18, 22]. Analysis of the absolute reliability during performance-related tests in non-athletic settings demonstrated that CV% below 10% are regarded as acceptable agreement [17], while Fox et al. [16] specified the threshold of acceptable reliability as not more than 10% of SEM.

In this regard, the balance beam walking test showed poor absolute reliability in boys, girls, and all the subjects. Further evaluation of the results in different age subgroups (Table 3) demonstrated that several measurements, including 10-m SRT (3.5 to 5.5-year-old subjects), DTH (4.5-year-old subjects), and balance beam walking test (4-, 5-, and 5.5-year-old subjects), failed to reach a satisfactory relative reliability level. Notably, SLJ, TT, and sit-and-reach tests that primarily measured the distance rather than the time can produce better intertrial relative reliability results in preschoolers. This finding may be related to the nature and complexity of required motor skills in measurements. Furthermore, the balance beam walking test for all the subdivided age groups and the TT test for 3.5-year-old subjects showed an unacceptable level of absolute reliability.

Recent studies have reported that the complexity of tests directly alters the consistency of their testing results [23–25]. Only a limited or short distance of locomotion was required in the sit-and-reach, TT, and SLJ tests of our study. Conversely, 10-m SRT, DTH, and balance beam walking test required preschoolers to walk, run, or jump over remarkably longer distances and testing durations. Therefore, these measurements included additional repeated movements and potentially high demands on movement consistency. Moreover, subjects can start their test with preplanned or preprogrammed motor skills (open-loop control-oriented items) without the stress of time limits during sit-and-reach, TT, and SLJ tests. Conversely, 10-m SRT, DTH, and balance beam walking test required subjects to execute motor skills using closed-loop control and integrate sensory feedback for movement or postural corrections during processes [26]. Therefore, subsequent repeated movements must be completed continuously without pause or other preparation time once these tests have started. Testing items that use closed-loop motor control can potentially lead to increased inconsistency in the testing results and hence relatively poor test–retest reliability in preschool children.

Apart from issues of test characteristics, previous studies showed that older preschool children demonstrate superior motor performance in both locomotion and object control [11, 12]. The comparison of relative and absolute reliability in our study clearly demonstrated that the oldest group (6-year-old subjects) generally showed a higher degree of relative and absolute reliability than the youngest group (3-year-old subjects). Gabbard [27] recently reported that refinement and maturation of fundamental motor skills only occur during late childhood (age of 6–12 years). Latorre Román et al. [5] presented high consistency of motor performance in the same testing items among older preschool children; therefore, maturity of preschool children can be a key factor that affects intraday reliability [5].

Furthermore, 10-m SRT, sit-and-reach, and balance beam walking tests are more reliable when preschool girls are tested, while TT and DTH are more reliable when preschool boys are examined. Although preschool boys and girls showed similar object control and locomotor skills in some studies [28, 29], Hardy et al. [30] found that girls performed better than boys in locomotor skills. Regarding balance performance, girls demonstrated better postural control and hence superior performance in balance tasks than boys [31–33]. Previous studies also showed that girls outperformed boys in flexibility throughout childhood until adolescence [3, 32]. The comparison of TT ability exhibited consistent results with a previous study such that superior performance in male children was observed [34]. Although investigations on DTH are lacking, recent studies indicated that boys performed significantly better than girls in leap, SLJ, and sideway jump tests [35, 36]. These findings are consistent with our study, wherein boys showed better performance in both relative and absolute reliability in DTH. The

                                                      

improved intertrial relative reliability of certain testing items of genders may be explained via two aspects. (1) The more superior motor skills and development demonstrated by boys or girls in certain testing items can lead to both high and consistent motor performance. (2) The learning effect available for skills that are already well performed is related to the diminishing gain or decreased margins.

Apart from the relative and absolute reliability, estimating the MDC with 95% confidence interval ($MDC_{95}$) was recommended in recent studies [20]. Determining whether the observed change is due to the real effect from intervention or measurement error is unclear without prior knowledge of the MDC value although a high degree of test–retest reliability is provided. Our results demonstrated very large $MDC_{95}$ values for all subjects in the balance beam walking test at 4.09 s, which is 54.9% of the performance of the better trial (7.45 s). Hence, preschool children must achieve a reduction of at least 55% in their balance beam walking time to show meaningful or real improvement with 95% confidence for excluding errors induced during the measurement. In this regard, further investigations on the source of measurement errors or reasons for such unreliable performance during the balance beam walking test for preschool children are necessary. Otherwise, the government should consider devising another test to replace the balance beam walking assessment and produce improved reliability and usefulness and valid results for testing dynamic balance.

Apart from reliability data and $MDC_{95}$ values, practitioners also intend to determine threshold values beyond zero that can represent the minimum change required for practically meaningful results using SWC. SWC and SEM values are commonly compared to express and understand test sensitivity [17]. Briefly, Liow and Hopkins [37] established thresholds to determine whether a test has "good sensitivity" and detect changes if SEM is smaller than SWC; the test has "satisfactory sensitivity" if SEM is equal to SWC, while the test only has "marginal sensitivity" if SEM is larger than SWC. The analysis of NPFM sensitivity exhibited that the effectiveness of each testing item in NPFM to detect real and practically meaningful change in the performance of individuals can be verified. The sit-and-reach test in our study showed good sensitivity in all the groups, except for 3.5- and 4-year-old subjects. Despite the gender and age consideration, SWC of the sit-and-reach test for all the preschool children was 0.90, while SEM and $MDC_{95}$ were 0.63 and 1.74 cm, respectively. Therefore, any observed change beyond 0.90 cm can be regarded as practically meaningful and exceeds the typical error of measurement. Practitioners have 95% confidence to consider the change as real rather than a measurement error when the observed change is over 1.74 cm. By comparison, SLJ only showed good sensitivity when it was used in the group of all subjects, boys, and girls but only marginal sensitivity was observed in all the subdivided age groups. Similarly, the TT test only showed good sensitivity in boys and 4.5-year-old subjects and satisfactory sensitivity in overall and 4-year-old subjects. Moreover, 10-m SRT, DTH, and balance beam walking test showed marginal sensitivity in most groups. Among the testing items of NPFM, only SLJ, TT, and sit-and-reach tests were considered simple tests using open-loop control and showed good or satisfactory sensitivity in several subject groups. Therefore, typical errors with relatively low SEM and high SWC values in these three testing items will unlikely mask the detectable and meaningful improvement when used in particular preschool groups [38].

Paired sample t-test revealed significant differences between trials while clear improvements with moderate to large effects were observed on the second trial of all the tests, thereby showing considerable systematic bias. Given that original NPFM guidelines require preschool children to remain resting and avoid unnecessary vigorous activities before conducting testing items, relevant information regarding warm-up or familiarization sessions is unavailable. Our study only provided instructions and demonstrations to reflect the actual reliability and sensitivity performance of NPFM and conform with the current NPFM guidelines. In this regard,

previous studies reported that the induced residual learning effect can reach 60 days [39, 40]. A recent study showed that motor test performance in preschool children peaked at the fourth or fifth session [41]. Therefore, the clear improvement of our second trial may be related to the carryover learning or warm-up effect induced from the first trial, especially when preschoolers were not fully familiar with the performance of motor tasks. Tomac and Hraski [41] recommended using five trials for each testing item for preschool children to remove the potential learning effect from the first few attempts without provoking transformational effects. Therefore, practitioners and researchers of future studies should provide at least four and optimally five relevant familiarization sessions before using NPFM when conducting fitness tests on preschool children, with each test having five trials to maximize the consistency. Although our study did not compare differences between tests with or without warm-up sessions, a standardized pretest warm-up protocol should be added in NPFM guidelines and implemented in the future for both safety and performance reasons. A simple pretest warm-up protocol for preschoolers adopted in a recent study can be directly referenced or used with proper modification, including five minutes of low-intensity running, followed by another five minutes of general exercises, such as skipping, leg lifts, lateral running, and front-to-behind arm rotations, to cover all body regions and simulate movements of testing items in NPFM [5].

The results of this study provided researchers and preschool teachers empirical evidence regarding the test–retest reliability of measurements in NPFM. The provided SWC and $MDC_{95}$ values can give practitioners concrete information regarding minimum differences required to reflect true performance changes. However, limitations still exist in this study. First, older preschoolers will have relatively more experience in performing testing items than younger groups, which had insufficient pretest familiarization sessions, because NPFM is conducted on preschool children on a yearly basis. Second, learning or practicing effects were very likely induced during the initial trial of most testing items due to our strict adherence to the original NPFM protocol of not requiring any warm-up or familiarization period. Third, learning or practicing effects induced in each group can vary due to gender, age, and maturity differences among subjects. Finally, the 3-year-old group was investigated because the timing of our study mismatched the academic year.

In conclusion, all the six measurement items in NPFM provided good relative reliability when conducted on the same day with repeated measures. The balance beam walking test showed low absolute reliability (>10%) in both SEM% and CV%. Systematic bias was observed with significantly improved performance during the second trial of all the tests.

## Supporting information

**S1 Data.**
(XLSX)

## Acknowledgments

The authors would like to thank the subjects for their participation in this research. Working with so many preschool children is such a great and unforgettable experience. We express our appreciation to the parents and kindergarten teachers of the subjects as well as graduate students from Beijing Sports University for their assistance and support.

## Author Contributions

**Conceptualization:** Indy Man Kit Ho.

**Data curation:** Indy Man Kit Ho.

**Investigation:** Hua Fang.

**Project administration:** Hua Fang.

**Resources:** Hua Fang.

**Supervision:** Indy Man Kit Ho.

**Writing – original draft:** Hua Fang.

**Writing – review & editing:** Indy Man Kit Ho.

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
