## [Decision Letter · Decision Letter 0]

7 Sep 2020

PONE-D-20-20616

The intraday reliability, sensitivity and minimum detectable change of National Physical Fitness Measurement for Preschool Children in China

PLOS ONE

Dear Dr. Ho,

Thank you for submitting your manuscript to PLOS ONE. After careful consideration, we feel that it has merit but does not fully meet PLOS ONE’s publication criteria as it currently stands. Therefore, we invite you to submit a revised version of the manuscript that addresses the points raised during the review process.

We look forward to receiving your revised manuscript.

Kind regards,

Subas Neupane

Academic Editor

PLOS ONE

Journal Requirements:

Additional Editor Comments (if provided):

Two reviewers have provided comments on your manuscript. Both the reviewers have good points, please consider revising the manuscript addressing each of the comments raised. Beside that, the English language of the manuscript should be checked. Another minor issue is, in results section the sub-headings are too long, please make them short using only the relevant text.

Reviewers' comments:

Reviewer's Responses to Questions

**Comments to the Author**

1. Is the manuscript technically sound, and do the data support the conclusions?

Reviewer #1: Yes

Reviewer #2: Partly

2. Has the statistical analysis been performed appropriately and rigorously? 

Reviewer #1: Yes

Reviewer #2: No

3. Have the authors made all data underlying the findings in their manuscript fully available?

Reviewer #1: Yes

Reviewer #2: No

4. Is the manuscript presented in an intelligible fashion and written in standard English?

Reviewer #1: Yes

Reviewer #2: No

5. Review Comments to the Author

Reviewer #1: Thank you for inviting me to review the manuscript on “The intraday reliability, sensitivity and minimum detectable change of National Physical Fitness Measurement for Preschool Children in China”. This research investigated the reliability, sensitivity and minimum detectable change values of NPFM in preschool children aged between 3.5 to 6 years. Overall the manuscript is well-written. I have a few minor comments that I would like the authors to consider.

1. In Abstract, Please provide more detail on the reliability and sensitivity analysis method.

2. Line 32, the keywords of “muscle strength; balance test” should be changed, ie. test-retest reliability...

3. In the Procedures, please clarify no previous familiarization was given for any test, although it was mentioned in line 342.

how much studies were identified in the review and how many provided data. Are there any differences in design or populations of those who provide data versus those who do not?

4. I feel confused there were two age groups (3.5-year, 4-year and 4.5-year vs. 5-year, 5.5-year and 6-year) in analyzing the intraday reliability, SEM, SWC, MDC95 and classification of sensitivity, on page 12 lines 208-237, why showed the results in table 3 & table 4 ?

5. I’m also confused the sentence, “To further improve the test-retest reliability of NPFM in preschoolers of different age groups or genders, researchers and practitioners should provide sufficient warm-up and practice opportunity to minimize learning effects”, What is meant by it. Which results could be deduced such the conclusion or advice in this manuscript?

Reviewer #2: Line 17 : please change "National Physical Fitness Measurement (preschool children version)” to “National Physical Fitness Measurement (NPFM - preschool children version)”

Line 23: please mention the model of ICC that you used

Line 23: Change “(ICC = 0.77 to 0.97)" to “(ICC…: 0.77 to 0.97)"

Line 24: Change “(moderate: ICC = 0.56 to 0.74)” to “(ICC: 0.56 to 0.74 [moderate])”

Line 25-26: Change “subject (poor: ICC 0.33 to 0.35), 5.5-year subject (moderate: ICCs=0.68) and double-leg timed hop test (moderate: ICC = 0.67) in 4.5-year.” to “subject (ICC: 0.33 to 0.35 [poor]), 5.5-year subject (ICC=0.68 [moderate]) and double-leg timed hop test (ICC = 0.67 [moderate]) in 4.5-year.”

Line 26-27: based on which results/statistical index?? What about the absolute reliability results?

Line 28-31: try to generalize your conclusion not a simple repetition of results.

Line 58-72: It is a classic description of reliability and sensitivity statistic tools, so please move this paragraph to discussion section or remove it. It should be better to highlight the meaning and the importance of the absolute and relative reliability and the internal, external sensitivity.

Line 99-100: please edit the form to “3.5≤ (n= 31)<4 years-old, 4≤ (n = 22) <4.5-years-old…….”

Table 1: please insert the seize of each group, for example change “All ages” to “All ages (n=209)

Line 113: it a simple randomization or counterbalanced?

Line 114: without familiarization session?

Line 113-116: indoor or outdoor? At the same time of day?

Line 115: As a general testing instruction for young children, it must do more than trial for each test.

Line 140-148: for 10-meter shuttle run test and Balance beam walking tests, are you sur that subjects have a complete recovery after only 1 min of rest?

Line 181-189: Are you checked to normality of data distribution? I think so that you don’t need to apply a log transformation with data normally distributed, and also with medium sample seize (greater than 20), it is recommended to combine t-student with effect seize Cohen d than use a non-clinical magnitude-based inference statistics.

Lines 195-198, 213-216 and 231-236: you focus only to interpret the ICC results, what about SEM and MDC values? For example, if MDC95 of 10m Shuttle run (s) equal to 1.01, how interpret this result?? Same for SEM values

Table 3 and Table 4: please combine table 3 with table 4

Tables 2-4: You mentioned a P signification values, but you don’t mentioned which statistical tool that you used?

Line 255-257: Based only ICC results you cannot conclude that tests has a good reliability.

Line 320-324: General interpretation with lack of explanation of the meaning and the exact utility of SEM and SWC.

Line 323: Which type of “detect true changes”?

Line 331-334: there is a lack of warm-up protocol description.

6. PLOS authors have the option to publish the peer review history of their article (what does this mean?). If published, this will include your full peer review and any attached files.

Reviewer #1: No

Reviewer #2: **Yes: **Wissem Dhahbi

---

## [Author Response · Author response to Decision Letter 0]

2 Oct 2020

We thank the two reviewers for their time and valuable suggestions. In this revised version we have answered all the questions raised by the reviewers and edited the manuscript with substantial revision on English syntax accordingly. We hope that this revised manuscript meets the standard for publication in PLOS ONE. Below please find our point-to-point responses to reviewers.

Reviewer: 1

1. In Abstract, Please provide more detail on the reliability and sensitivity analysis method.

Authors’ reply: According to the reviewers’ suggestion, abstract was revised as “Intraday relative reliability was tested using intraclass correlation coefficient (ICC3,1) with a 95% confidence interval while absolute reliability was expressed in standard error of measurement and percentage of coefficient of variation (CV%). Test sensitivity was assessed by comparing the smallest worthwhile change (SWC) with standard error of measurement (SEM), while MDC values with 95% confidence interval (MDC95) were established.” (Line 22-27). In addition, more contents regarding SEM and SWC for showing the sensitivity result in “The balance beam walking test showed poor absolute reliability in all the groups (SEM%: 11.76 to 22.28 and CV%: 15.40 to 24.78). Both standing long jump and sit-and-reach tests demonstrated good sensitivity (SWC > SEM) in all subjects group, boys, and girls.” as shown (Line 32-35).

2. Line 32, the keywords of “muscle strength; balance test” should be changed, ie. test-retest reliability...

Authors’ reply: According to the reviewers’ suggestion, keywords “muscle strength; balance test” had been changed to “test-retest reliability” (Line 42)

3. In the Procedures, please clarify no previous familiarization was given for any test, although it was mentioned in line 342. According to the reviewer’s suggestion, relevant content was added as “According to the current NPFM guidelines [8], no previous familiarization session was given.” (Line115-116)

how much studies were identified in the review and how many provided data. Are there any differences in design or populations of those who provide data versus those who do not?

Authors’ reply: In our literature review, about 12 papers regarding fitness battery/protocol for children were identified. Reference [1-3, 11,32] was more about youth, adolescents or children but not preschoolers. [4] by Ortega et al. (2014) provided systematic review regarding the field-based physical fitness-test battery for preschool children using PREFIT battery. [5-7] were some cross-sectional study testing physical fitness in preschool children in Spain, Colombia and China. [9] was the reliability and feasibility study using PREFIT battery in Spain. [12] was about the motor skill performance (not purely physical fitness test battery) on 3- and 4-year old preschoolers in America. [33] provided preliminary performance results of PREFIT battery conducted in Preschool children (aged 3.00 to 6.25) in Spain with percentiles classified.

The systematic review by Ortega et al. (2014) provided most comprehensive information regarding reliability. It has reported 21 relevant articles examining reliability. For example, some studies cited in their paper showed reliable results using 1/2-mile walk/run test to assess 5 years old preschoolers with (r > 0.73) while another study by Niederer et al. (cited in Ortega’s article) showed good reliability of 20-m shuttle-run test in Swiss preschool children aged 4-6 years (r = 0.84). However all these studies only covered preschoolers aged 4-6 years but not those below 4. Apart from the r value reflecting the correlation coefficient, another most commonly used reliability measure was ICC. Ortega et al. also reported the result regarding standing long jump from other articles that Krombholz (2011) observed r=0.68 in using standing long jump for those 3- to 7-year-old but the results were obtained with 8 months apart. Meanwhile Ortega also reported another standing-long jump result using ICC (0.65-0.89) in 4 to 5 year old children showing acceptable ICC. In addition, Ortega et al. also reported the reliable results from one-leg-stance (ICC: 0.73 to 0.99; r = 0.84-0.97) in preschool children of different ages from several other papers. They also cited another paper from Oja and Jurimae (1997) to show acceptable reliabilities of 4 x 10 m shuttle run using ICC and Cronbach’s alpha for boys and girls aged 4 to 5 years.

Apart from Ortega et al. Amado-Pacheco et al. (2019) performed Fuprecol kids study with 90 preschool children between 3 to 5 years old using inter-day comparison approach (two testing sessions with two weeks apart). They performed a well-known PREFIT 20 m shuttle run test for cardiorespiratory performance, standing long jump and handgrip for strength and musculoskeletal performance, 4 x 10 m shuttle run for speed and agility and sit and reach for flexibility. This research group assess reliability by using mean differences comparison, ICC values, Bland-Altman plots and technical error of measurement (TEM). They have showed -0.27 cm of boys between trials in sit-and-reach and 0.59 cm (p<0.01) increase of performance in girls. In general they reported excellent ICC values for standing broad jump (0.99), 4 x 10m shuttle run (0.95) and sit and reach (0.96).

In addition, another study conducted by Cadenas-Sanchez et al. (2016) also used inter-day comparison (2 weeks apart) to assess 161 Spanish preschoolers aged 3 to 5 years with PREFIT 20 m shuttle run, handgrip strength, standing long jump, 4 x 10 m shuttle run and one-leg-stance tests. They used Bland-Alman method and paired sample t-test to check if error was significantly different from reference point. They have shown significantly shorter standing long jump distance but longer one-leg stance performance between days.

Therefore, the methods used for assessing reliability were mixed mainly including correlation coefficient, ICC and Bland-Altman plots while most widely studied fitness test battery was PREFIT in European regions. Most studies focused on 3 to 5 years old or 4 to 6 years old and therefore, data of the entire spectrum (3 to 6 years) in preschoolers might not be complete in each study. 

We have mildly revised the introduction section by adding a brief summary of current review to also make the transition of sentences/paragraphs more smooth. “Previous studies showed excellent reliability of FITness testing in PREschool children (PREFIT) in Spain using Bland-Altman method, intra-class correlation coefficient (ICC) and the comparison of mean differences [6, 9]. Meanwhile, the systematic review from Ortega et al. [4] reported that 4 x 10 m shuttle-run test has provides reliable measures in speed and agility related fitness for preschoolers aged 4 to 5 years (ICC: 0.52 to 0.92) and one-leg-stance test is a popular and reliable test for assessing the balance of preschool children (ICC: 0.73 to 0.99). In addition, the standing long jump test used in testing 4- and 5-year-old preschool children showed acceptable relative reliability (ICC: 0.65 to 0.89). Regarding the studies using Chinese NPFM, the level of physical fitness and activity of preschool children in Shanghai was reported recently [7, 10].” Line (69-79)

4. I feel confused there were two age groups (3.5-year, 4-year and 4.5-year vs. 5-year, 5.5-year and 6-year) in analyzing the intraday reliability, SEM, SWC, MDC95 and classification of sensitivity, on page 12 lines 208-237, why showed the results in table 3 & table 4 ?

Authors’ reply: We agreed with the suggestions from both reviewers as the key of this paper was not to compare the differences between younger and older preschoolers. Now Table 3 and table 4 are combined together as Table 3 only now (Line 240-243)

5. I’m also confused the sentence, “To further improve the test-retest reliability of NPFM in preschoolers of different age groups or genders, researchers and practitioners should provide sufficient warm-up and practice opportunity to minimize learning effects”, What is meant by it. Which results could be deduced such the conclusion or advice in this manuscript?

Yes, we agree that the current study did not directly investigate the warm up effect. Whereas per the request from another reviewer, paired sample t-test was used to better show the existence of systematic bias. In the discussion, although we proposed the observed significant improvement in the 2nd trial could be induced by learning or warm-up effect (Line 397), to avoid such potential confusion you mentioned, we added contents to clarify that a standardized warm up protocol should be warranted for both performance and safety reasons as “Although our study did not compare differences between tests with or without warm-up sessions, a standardized pretest warm-up protocol should be added in NPFM guidelines and implemented in the future for both safety and performance reasons. A simple pretest warm-up protocol for preschoolers adopted in a recent study can be directly referenced or used with proper modification, including five minutes of low-intensity running, followed by another five minutes of general exercises, such as skipping, leg lifts, lateral running, and front-to-behind arm rotations, to cover all body regions and simulate movements of testing items in NPFM [5].” Line (409-417)

Reviewer: 2

Line 17 : please change "National Physical Fitness Measurement (preschool children version)” to “National Physical Fitness Measurement (NPFM - preschool children version)”

Authors’ reply: We have amended the abstract according to the suggestion. The revised content is “China General Administration of Sport has published and adopted the National Physical Fitness Measurement (NPFM - preschool children version) since 2000.” shown (Line 16-17)

Line 23: please mention the model of ICC that you used

Authors’ reply: We have added the adopted model of ICC back to the content of abstract as “Intraday relative reliability was tested using intraclass correlation coefficient (ICC3,1) with a 95% confidence interval while absolute reliability was expressed in standard error of measurement and percentage of coefficient of variation (CV%).” (Line 22-24). It was also shown clearly in the abstract throughout as “Measurements in most groups, except 10-m shuttle run test (ICC3,1: 0.56 to 0.74 [moderate]) in the 3.5 to 5.5-year-old groups, balance beam test in 4- and 5-year-old (ICC3,1: 0.33 to 0.35 [poor]) and 5.5-year-old (ICC3,1=0.68 [moderate]) groups, and double-leg timed hop test (ICC3,1=0.67 [moderate]) in the 4.5-year-old group, demonstrated good to excellent relative reliability (ICC3,1: 0.77 to 0.97).” (Line 27-32)

Line 23: Change “(ICC = 0.77 to 0.97)" to “(ICC…: 0.77 to 0.97)"

Authors’ reply: According to the reviewers’ suggestion “(ICC = 0.77 to 0.97)” was changed to “(ICC3.1: 0.77 to 0.97)” (Line 32)

Line 24: Change “(moderate: ICC = 0.56 to 0.74)” to “(ICC: 0.56 to 0.74 [moderate])”

Authors’ reply: According to the reviewers’ suggestion “(moderate: ICC = 0.56 to 0.74)” was changed to “(ICC3.1: 0.56 to 0.74 [moderate])” (Line 28 )

Line 25-26: Change “subject (poor: ICC 0.33 to 0.35), 5.5-year subject (moderate: ICCs=0.68) and double-leg timed hop test (moderate: ICC = 0.67) in 4.5-year.” to “subject (ICC: 0.33 to 0.35 [poor]), 5.5-year subject (ICC=0.68 [moderate]) and double-leg timed hop test (ICC = 0.67 [moderate]) in 4.5-year.”

Authors’ reply: According to the reviewers’ suggestion “subject (poor: ICC 0.33 to 0.35), 5.5-year subject (moderate: ICCs=0.68) and double-leg timed hop test (moderate: ICC = 0.67) in 4.5-year.” was changed to “Measurements in most groups, except 10-m shuttle run test (ICC3,1: 0.56 to 0.74 [moderate]) in the 3.5 to 5.5-year-old groups, balance beam test in 4- and 5-year-old (ICC3,1: 0.33 to 0.35 [poor]) and 5.5-year-old (ICC3,1=0.68 [moderate]) groups, and double-leg timed hop test (ICC3,1=0.67 [moderate]) in the 4.5-year-old group, demonstrated good to excellent relative reliability (ICC3,1: 0.77 to 0.97).” (Line 27-32 )

Line 26-27: based on which results/statistical index?? What about the absolute reliability results?

Authors’ reply: To better clarify, the sentence is now revised as “Both standing long jump and sit-and-reach tests demonstrated good sensitivity (SWC > SEM) in all subjects group, boys, and girls.” Line (33-35). Meanwhile, the absolute reliability in terms of SEM% and CV% of the worst testing item was also highlighted as “The balance beam walking test showed poor absolute reliability in all the groups (SEM%: 11.76 to 22.28 and CV%: 15.40 to 24.78).” Line (32-33).

Line 28-31: try to generalize your conclusion not a simple repetition of results.

Authors’ reply: We based on the result of the use of pairwise comparison showed systematic bias between trials and also the newly added discussion part concerning the recommended number of familiarization sessions and testing trials to revise the conclusion in a more concrete and specific approach as “Pairwise comparison revealed systematic bias with significantly better performance in the second trial (p<0.01) of all the tests with moderate to large effect size. Hence, NPFM guidelines should be revised by adding adequate familiarization sessions and standardized warm-up protocols as well as increasing the number of testing trials. SWC and MDC95 values of NPFM tests should be considered to realize true performance changes.” Line (35-40)

Line 58-72: It is a classic description of reliability and sensitivity statistic tools, so please move this paragraph to discussion section or remove it. It should be better to highlight the meaning and the importance of the absolute and relative reliability and the internal, external sensitivity.

Authors’ reply: According to the suggestion of reviewer, we have moved the contents of those sentences back to discussion. In the initial part of the discussion, we have explained the possible sources leading to systematic bias as “This study primarily aimed to set up the intraday reliability, MDC, and sensitivity of six key testing items of NPFM by comparing between trials. The systematic bias of observed differences, such as potential of the learning effect to lead to a higher degree of familiarity of the selected measurement, insufficient recovery from the previous trial that induces the fatigue effect to subsequent attempts, and different emotional statuses or motivation levels, can be detected when intertrial reliability is determined [17].” (Line 263-268). Meanwhile, the original contents existed in the introduction part was further revised by explaining the limitation of ICC and introducing the use of absolute reliability as “ICC is commonly used to assess the reliability of a measurement or testing method, wherein values over 0.90 are regarded as excellent relative test–retest reliability. Tests with excellent ICCs exhibit good stability and consistency of measurement over time and low measurement error [20]. However, previous studies reported limitations, such as inter subject variability that can potentially affect the result and overestimated ICC values in a typically heterogeneous population, in the use of ICC alone [21]. Therefore, measurements with excellent relative reliability do not necessarily ensure consistent intertrial performance. Calculations of SEM and CV% were further recommended to obtain within-subject variation in addition to measuring ICCs and confirm the absolute reliability [18, 22]. Analysis of the absolute reliability during performance-related tests in nonathletic settings demonstrated that CV% below 10% are regarded as acceptable agreement [17], while Fox et al. [16] specified the threshold of acceptable reliability as not more than 10% of SEM.” Line (272-284). To better elaborate the importance of MDC95 and SWC, the discussion part was revised by using our findings to explain the interpretation of MDC and SWC as “Apart from the relative and absolute reliability, estimating the MDC with 95% confidence interval (MDC95) was recommended in recent studies [20]. Determining whether the observed change is due to the real effect from intervention or measurement error is unclear without prior knowledge of the MDC value although a high degree of test–retest reliability is provided. Our results demonstrated very large MDC95 values for all subjects in the balance beam walking test at 4.09 s, which is 54.9% of the performance of the better trial (7.45 s). Hence, preschool children must achieve a reduction of at least 55% in their balance beam walking time to show meaningful or real improvement with 95% confidence for excluding errors induced during the measurement.” Line (349-358). In addition, a paragraph was added to elaborate the use of SWC and SEM values for acquiring the sensitivity as “Apart from reliability data and MDC95 values, practitioners also intend to determine threshold values beyond zero that can represent the minimum change required for practically meaningful results using SWC. SWC and SEM values are commonly compared to express and understand test sensitivity [17]. Briefly, Liow and Hopkins [37] established thresholds to determine whether a test has “good sensitivity” and detect changes if SEM is smaller than SWC; the test has “satisfactory sensitivity” if SEM is equal to SWC, while the test only has “marginal sensitivity” if SEM is larger than SWC. The analysis of NPFM sensitivity exhibited that the effectiveness of each testing item in NPFM to detect real and practically meaningful change in the performance of individuals can be verified.” Line (364-373). Through the elaboration from our findings, reader can explicitly know the importance of a sensitive test through “Despite the gender and age consideration, SWC of the sit-and-reach test for all the preschool children was 0.90, while SEM and MDC95 were 0.63 and 1.74 cm, respectively. Therefore, any observed change beyond 0.90 cm can be regarded as practically meaningful and exceeds the typical error of measurement. Practitioners have 95% confidence to consider the change as real rather than a measurement error when the observed change is over 1.74 cm. By comparison, SLJ only showed good sensitivity when it was used in the group of all subjects, boys, and girls but only marginal sensitivity was observed in all the subdivided age groups. Similarly, the TT test only showed good sensitivity in boys and 4.5-year-old subjects and satisfactory sensitivity in overall and 4-year-old subjects. Moreover, 10-m SRT, DTH, and balance beam walking test showed marginal sensitivity in most groups.” Line (374-385)

Line 99-100: please edit the form to “3.5≤ (n= 31)<4 years-old, 4≤ (n = 22) <4.5-years-old…….”

Authors’ reply: amended per suggestion as “Subjects were further divided into the following subgroups according to their chronological ages: ≤ 3.5 (n=31) < 4, ≤ 4 (n=22) < 4.5, ≤ 4.5 (n=43) < 5, ≤ 5 (n=24) < 5.5, ≤ 5.5 (n=45) < 6, and ≤ 6 (n=44) years old.” (Line 98-100)

Table 1: please insert the seize of each group, for example change “All ages” to “All ages (n=209)

Authors’ reply: We have amended per suggestion as shown in Table 1 (Line 110). To avoid confusing readers “all ages” as all six subgroups by different ages, we have changed “all age” to “all subjects”

Line 113: it a simple randomization or counterbalanced?

Authors’ reply: Since we had 6 different testing items that could produce 720 possible sequence. A complete/ideal counterbalanced is not possible and therefore we have adopted randomization (mentioned in line 115)

Line 114: without familiarization session?

Authors’ reply: Authors fully understand the importance of using familiarization session to enhance the test retest reliability. However, to truly reflect the current practice of NPFM adopted in China, we strictly followed the protocol such that we could based on any observed systematic bias to make suggestion in our discussion section. Per the request of another reviewer, the relevant sentence was added as “According to the current NPFM guidelines [8], no previous familiarization session was given.” Line (115-116). Meanwhile, in our discussion we have added a paragraph to make use the recent paper from Tomac and Hraski [41] to counter propose the need of using multiple familiarization sessions as “Given that original NPFM guidelines require preschool children to remain resting and avoid unnecessary vigorous activities before conducting testing items, relevant information regarding warm-up or familiarization sessions is unavailable. Our study only provided instructions and demonstrations to reflect the actual reliability and sensitivity performance of NPFM and conform with the current NPFM guidelines. In this regard, previous studies reported that the induced residual learning effect can reach 60 days [39, 40]. A recent study showed that motor test performance in preschool children peaked at the fourth or fifth session [41]. Therefore, the clear improvement of our second trial may be related to the carryover learning or warm-up effect induced from the first trial, especially when preschoolers were not fully familiar with the performance of motor tasks. Tomac and Hraski [41] recommended using five trials for each testing item for preschool children to remove the potential learning effect from the first few attempts without provoking transformational effects. Therefore, practitioners and researchers of future studies should provide at least four and optimally five relevant familiarization sessions before using NPFM when conducting fitness tests on preschool children, with each test having five trials to maximize the consistency.” Line (393-409).

Line 113-116: indoor or outdoor? At the same time of day?

Authors’ reply: We have supplemented relevant information in the procedures section as “NPFM was conducted by trained research assistants on a synthetic rubber surface at the outdoor playground of a kindergarten school in Beijing in the morning.” Line (113-114)

Line 115: As a general testing instruction for young children, it must do more than trial for each test.

Authors’ reply: Similar to the concern of not using familiarization session, authors understand the importance of multiple trials for getting a more steady and reliable results. However, the current NPFM stipulates the use of two trials for each testing item. Therefore, our study by strictly following the current practice of NPFM is a good opportunity to reflect the potential weakness of the current testing protocols. Therefore, in our discussion we have a paragraph to counter propose the need of using optimally five trials instead of two to enhance the reliability. “Tomac and Hraski [41] recommended using five trials for each testing item for preschool children to remove the potential learning effect from the first few attempts without provoking transformational effects. Therefore, practitioners and researchers of future studies should provide at least four and optimally five relevant familiarization sessions before using NPFM when conducting fitness tests on preschool children, with each test having five trials to maximize the consistency.” Line (403-409). We hope these can help clarify that it was not our originally intention to conduct any sub-quality/optimal fitness test whereas the primary aim of this paper was to reflect what the current NPFM looks like so that we can make recommendation to relevant organizations.

Line 140-148: for 10-meter shuttle run test and Balance beam walking tests, are you sur that subjects have a complete recovery after only 1 min of rest?

Authors’ reply: For explosive strength or power related test, we know that at least 1:10 or even 1:15 work-rest ratio is required to warrant complete recovery. For the Balance beam walking, the walking speed was much slower than normal walking which should have no potential issue regarding complete recovery. For the 10-meter shuttle run test, we have made literature search for similar motor or fitness test in assessing the agility for preschool children. Interestingly, only “Martinez-tellez et al., (2015). Health-related physical fitness is associated with total and central body fat in preschool children aged 3 to 5 years. Pediatric Obesity, 11(6), 468-474.” this study mentioned the use of 1-2 min rest between trials in 4x10 m shuttle run whereas all the later studies (which those we cited in our manuscript) referenced from this paper but all of them did not mention the inter-trial resting period. Since Martinez-tellez et al used 4x10 while our study using NPFM used 2x10 meters, to take reference from and in line with this existed resting standard for similar test on similar population, we used at least 1 minute (as the duration and distance was about half of those in Martinez-tellez et al.). Authors understand, the current recovery duration may not be 100% whereas, our pairwise comparison between 1st and 2nd trials did show improvement but not worse performance from incomplete recovery. Therefore, we believe the issue of incomplete recovery was minimum or negligible in the current study.

Line 181-189: Are you checked to normality of data distribution? I think so that you don’t need to apply a log transformation with data normally distributed, and also with medium sample seize (greater than 20), it is recommended to combine t-student with effect seize Cohen d than use a non-clinical magnitude-based inference statistics.

Authors’ reply: We have used Shapiro-Wilk test and qq plot to observe the normality. Although quite a number of groups violated the SW test, most groups in qq plot were on or very closed to the reference line with little or some deviation at the tail. Meanwhile, as all our groups have more than 20 sample size, we believe that we do not require using non-parametric methods or additional log-transformation to yield proper results. Per the suggestion from the review, we have changed the pairwise comparison from non-clinical MBI to paired sample t-test with Cohen’s d as effect size calculation as shown in “The results of pairwise sample t-test (Table 4) showed a significant difference between trials for all the measurements of the 10-m SRT (p<0.01 and d=0.87 [large]), SLJ (p<0.01 and d=0.71 [moderate]), TT (p<0.01 and d=0.84 [large]), DTH (p<0.01 and d=0.92 [large]), sit-and-reach (p<0.01 and d=1.57 [large]), and balance beam walking (p<0.01 and d=0.69 [moderate]) tests.” Line 251-258, Table 4.

Lines 195-198, 213-216 and 231-236: you focus only to interpret the ICC results, what about SEM and MDC values? For example, if MDC95 of 10m Shuttle run (s) equal to 1.01, how interpret this result?? Same for SEM values

Table 3 and Table 4: please combine table 3 with table 4

Authors’ reply: Per the request of both reviewers, tables 3 and 4 are combined to one single table 3 as shown in line 240-243 now. 

We agree that only reporting ICC (relative reliability) may lead to biased or limited interpretation and therefore, the result section was revised with added content while CV% was also expressed to further strengthen the absolute reliability. “Table 2 shows good to excellent ICCs (0.77 to 0.97) of all the measurements in the groups of all subjects, boys, and girls. However, the balance beam walking test demonstrated poor absolute reliability for the groups of all ages (SEM%=18.05 and CV%=20.43), boys (SEM%=17.96 and CV%=20.47), and girls (SEM%=18.10% and CV%=20.38%). MDC95 values in the balance beam walking test for groups of all subjects, boys, and girls showed a minimum threshold of 4.09, 3.99, and 4.18 s, respectively, which are beyond the random measurement error with a 95% confidence level. 

SLJ demonstrated good sensitivity in the group of all subjects (SWC=4.54 > SEM=3.81), boys (SWC=4.68 > SEM=3.94), and girls (SWC=4.33 > SEM=3.67). Similarly, the sit-and-reach test showed good sensitivity in the group of all subjects (SWC=0.90 > SEM=0.63), boys (SWC=0.77 > SEM=0.68), and girls (SWC=0.89 > SEM=0.41). Only the boys group (SWC=0.40 > SEM=0.30) exhibited good sensitivity in the TT test, while satisfactory sensitivity was observed in all the subjects (SWC=38 ≈ SEM=0.36).” Line (197-211). Similarly, the SEM, SWC and CV% were supplemented in “Intraday reliability in ICC, CV%, SEM, SWC, and MDC95 data and classification of sensitivity in 3.5-, 4-, 4.5-, 5-, 5.5-, and 6-year-old subjects are presented in Table 3. The majority of measurements showed good to excellent relative reliability (ICC: 0.79 to 0.95), except the 10-m SRT (ICC: 0.67 to 0.73 [moderate]) in three groups (3.5-, 4-, and 5-year-old subjects), balance beam test (ICC: 0.33 to 0.68 [poor to moderate]) in 4-, 5-, and 5.5-year-old subjects, and DTH (ICC=0.67 [moderate]) in 4.5-year-old subjects. However, according to SEM% and CV% values, the balance beam walking test demonstrated poor absolute reliability (SEM%: 11.25 to 22.28 and CV%: 15.40 to 24.78) for all the age groups.

The comparison of SWC and SEM values showed that most measurements demonstrated only marginal sensitivity, except the TT test of 4.5-year-old subjects (SWC=0.35 > SEM=0.30) and the sit-and-reach test of 4.5- (SWC=0.86 > SEM=0.43), 5- (SWC=1.05 > SEM=0.58), 5.5- (SWC=0.90 > SEM=0.56), and 6-year-old (SWC=1.00 > SEM=0.65) subjects. Meanwhile, satisfactory sensitivity was observed in the TT test of 4-year-old subjects (SWC=0.29 ≈ SEM=0.28) and DTH in 5- (SWC=0.58 ≈ SEM=0.55) and 6-year-old (SWC=0.21 ≈ SEM=0.23) subjects.” (Line 224-239). To further interpret and elaborate the SEM, MDC and SWC results, the section from line 278-284 provided the importance and threshold of absolute reliability as “Therefore, measurements with excellent relative reliability do not necessarily ensure consistent intertrial performance. Calculations of SEM and CV% were further recommended to obtain within-subject variation in addition to measuring ICCs and confirm the absolute reliability [18, 22]. Analysis of the absolute reliability during performance-related tests in nonathletic settings demonstrated that CV% below 10% are regarded as acceptable agreement [17], while Fox et al. [16] specified the threshold of acceptable reliability as not more than 10% of SEM.” Furthermore, the interpretation of large SEM and CV values regarded as poor absolute reliability for balance beam walking test was discussed in line 286 to 287 “In this regard, the balance beam walking test showed poor absolute reliability (SEM%: 17.96 to 18.10 and CV%: 20.38 to 20.47) in boys, girls, and all the subjects.”. Similarly, the unacceptable absolute reliability for Tennis Throwing for 3.5-year-old was shown in line 295 to 298 as “Furthermore, the balance beam walking test for all the subdivided age groups (SEM%: 11.25 to 22.28 and CV%: 15.40 to 24.78) and the TT test for 3.5-year-old subjects (SEM%=12.63 and CV%=17.81) showed an unacceptable level of absolute reliability. 

The interpretation of MDC and SWC as well as the sensitivity level were further elaborated from line 349-385 as “Apart from the relative and absolute reliability, estimating the MDC with 95% confidence interval (MDC95) was recommended in recent studies [20]. Determining whether the observed change is due to the real effect from intervention or measurement error is unclear without prior knowledge of the MDC value although a high degree of test–retest reliability is provided. Our results demonstrated very large MDC95 values for all subjects in the balance beam walking test at 4.09 s, which is 54.9% of the performance of the better trial (7.45 s). Hence, preschool children must achieve a reduction of at least 55% in their balance beam walking time to show meaningful or real improvement with 95% confidence for excluding errors induced during the measurement. In this regard, further investigations on the source of measurement errors or reasons for such unreliable performance during the balance beam walking test for preschool children are necessary. Otherwise, the government should consider devising another test to replace the balance beam walking assessment and produce improved reliability and usefulness and valid results for testing dynamic balance.

Apart from reliability data and MDC95 values, practitioners also intend to determine threshold values beyond zero that can represent the minimum change required for practically meaningful results using SWC. SWC and SEM values are commonly compared to express and understand test sensitivity [17]. Briefly, Liow and Hopkins [37] established thresholds to determine whether a test has “good sensitivity” and detect changes if SEM is smaller than SWC; the test has “satisfactory sensitivity” if SEM is equal to SWC, while the test only has “marginal sensitivity” if SEM is larger than SWC. The analysis of NPFM sensitivity exhibited that the effectiveness of each testing item in NPFM to detect real and practically meaningful change in the performance of individuals can be verified. The sit-and-reach test in our study showed good sensitivity in all the groups, except for 3.5- and 4-year-old subjects. Despite the gender and age consideration, SWC of the sit-and-reach test for all the preschool children was 0.90, while SEM and MDC95 were 0.63 and 1.74 cm, respectively. Therefore, any observed change beyond 0.90 cm can be regarded as practically meaningful and exceeds the typical error of measurement. Practitioners have 95% confidence to consider the change as real rather than a measurement error when the observed change is over 1.74 cm. By comparison, SLJ only showed good sensitivity when it was used in the group of all subjects, boys, and girls but only marginal sensitivity was observed in all the subdivided age groups. Similarly, the TT test only showed good sensitivity in boys and 4.5-year-old subjects and satisfactory sensitivity in overall and 4-year-old subjects. Moreover, 10-m SRT, DTH, and balance beam walking test showed marginal sensitivity in most groups.”

Tables 2-4: You mentioned a P signification values, but you don’t mentioned which statistical tool that you used?

Authors’ reply: We have removed all those p values as those were the significance of ICC values which are necessary and not the key interests of our study.

Line 255-257: Based only ICC results you cannot conclude that tests has a good reliability.

Authors’ reply: We have revised the sentence to “The findings shown in Table 2 indicated that all the testing items generally demonstrate a good to excellent relative reliability in preschool children (ICC: 0.77 to 0.97).” by adding “relative” (Line 270-272) to better clarify. In addition, the importance, threshold and values of absolute reliability using SEM and CV% were emphasized from Line 275-287 as “However, previous studies reported limitations, such as inter subject variability that can potentially affect the result and overestimated ICC values in a typically heterogeneous population, in the use of ICC alone [21]. Therefore, measurements with excellent relative reliability do not necessarily ensure consistent intertrial performance. Calculations of SEM and CV% were further recommended to obtain within-subject variation in addition to measuring ICCs and confirm the absolute reliability [18, 22]. Analysis of the absolute reliability during performance-related tests in nonathletic settings demonstrated that CV% below 10% are regarded as acceptable agreement [17], while Fox et al. [16] specified the threshold of acceptable reliability as not more than 10% of SEM. 

In this regard, the balance beam walking test showed poor absolute reliability (SEM%: 17.96 to 18.10 and CV%: 20.38 to 20.47) in boys, girls, and all the subjects. We believe these contents help to avoid any misleading to dogmatic conclusion. 

Line 320-324: General interpretation with lack of explanation of the meaning and the exact utility of SEM and SWC.

Authors’ reply: We have revised and added contents as “Apart from reliability data and MDC95 values, practitioners also intend to determine threshold values beyond zero that can represent the minimum change required for practically meaningful results using SWC. SWC and SEM values are commonly compared to express and understand test sensitivity [17]. Briefly, Liow and Hopkins [37] established thresholds to determine whether a test has “good sensitivity” and detect changes if SEM is smaller than SWC; the test has “satisfactory sensitivity” if SEM is equal to SWC, while the test only has “marginal sensitivity” if SEM is larger than SWC. The analysis of NPFM sensitivity exhibited that the effectiveness of each testing item in NPFM to detect real and practically meaningful change in the performance of individuals can be verified. The sit-and-reach test in our study showed good sensitivity in all the groups, except for 3.5- and 4-year-old subjects. Despite the gender and age consideration, SWC of the sit-and-reach test for all the preschool children was 0.90, while SEM and MDC95 were 0.63 and 1.74 cm, respectively. Therefore, any observed change beyond 0.90 cm can be regarded as practically meaningful and exceeds the typical error of measurement. Practitioners have 95% confidence to consider the change as real rather than a measurement error when the observed change is over 1.74 cm. By comparison, SLJ only showed good sensitivity when it was used in the group of all subjects, boys, and girls but only marginal sensitivity was observed in all the subdivided age groups. Similarly, the TT test only showed good sensitivity in boys and 4.5-year-old subjects and satisfactory sensitivity in overall and 4-year-old subjects. Moreover, 10-m SRT, DTH, and balance beam walking test showed marginal sensitivity in most groups. Among the testing items of NPFM, only SLJ, TT, and sit-and-reach tests were considered simple tests using open-loop control and showed good or satisfactory sensitivity in several subject groups. Therefore, typical errors with relatively low SEM and high SWC values in these three testing items will unlikely mask the detectable and meaningful improvement when used in particular preschool groups [38].” Line (364-389).

Line 323: Which type of “detect true changes”?

Authors’ reply: We have revised the whole paragraph as shown in the response in the previous question. Therefore, the confusing or vague wordings of “detect true changes” did not exist anymore. Instead we added the whole paragraph from line 364-389 to more explicitly explain and interpret why the sit-and-reach test was sensitive and useful using SEM, SWC and MDC values.

Line 331-334: there is a lack of warm-up protocol description.

Authors’ reply: Per the request of reviewer, we have referenced from a recent study regarding fitness/motor test for preschool children and suggested a structured and complete warm up protocol as “Although our study did not compare differences between tests with or without warm-up sessions, a standardized pretest warm-up protocol should be added in NPFM guidelines and implemented in the future for both safety and performance reasons. A simple pretest warm-up protocol for preschoolers adopted in a recent study can be directly referenced or used with proper modification, including five minutes of low-intensity running, followed by another five minutes of general exercises, such as skipping, leg lifts, lateral running, and front-to-behind arm rotations, to cover all body regions and simulate movements of testing items in NPFM [5].” Line (409-417)

---

## [Decision Letter · Decision Letter 1]

28 Oct 2020

PONE-D-20-20616R1

Intraday reliability, sensitivity and minimum detectable change of National Physical Fitness Measurement for Preschool Children in China

PLOS ONE

Dear Dr. Ho,

Thank you for submitting your manuscript to PLOS ONE. After careful consideration, we feel that it has merit but does not fully meet PLOS ONE’s publication criteria as it currently stands. Therefore, we invite you to submit a revised version of the manuscript that addresses the points raised during the review process.

ACADEMIC EDITOR:

In methods part of the abstract, please also mention, when the study was conducted.

Also, please state the Six items of NPFM in the methods.

In main text, please state what is the meaning of (ICC _3, 1_) as the readers may not be familiar with it.

The heading under results in the result section (line 193-195) and another one in line 220-22 are too long. Please make it short and clear.

In discussion, please do not repeat the statistical results.

The conclusion presented in the abstract and in the main texts should be aligned.

We look forward to receiving your revised manuscript.

Kind regards,

Subas Neupane

Academic Editor

PLOS ONE

Additional Editor Comments (if provided):

In methods part of abstract, please also mention, when the study was conducted.

Also, please state the Six items of NPFM in the methods.

In main text, please state what is the meaning of (ICC 3, 1) as the readers may not be familiar with it.

The heading under results in the result section (line 193-195) and another one in line 220-22 are too long. Please make it short and clear.

In discussion, please do not repeat the statistical results.

The conclusion presented in the abstract and in the main texts should be aligned.

Reviewers' comments:

Reviewer's Responses to Questions

**Comments to the Author**

1. If the authors have adequately addressed your comments raised in a previous round of review and you feel that this manuscript is now acceptable for publication, you may indicate that here to bypass the “Comments to the Author” section, enter your conflict of interest statement in the “Confidential to Editor” section, and submit your "Accept" recommendation.

Reviewer #1: All comments have been addressed

2. Is the manuscript technically sound, and do the data support the conclusions?

Reviewer #1: Yes

3. Has the statistical analysis been performed appropriately and rigorously? 

Reviewer #1: Yes

4. Have the authors made all data underlying the findings in their manuscript fully available?

Reviewer #1: Yes

5. Is the manuscript presented in an intelligible fashion and written in standard English?

Reviewer #1: Yes

6. Review Comments to the Author

Reviewer #1: I agree with Hraski and the author's suggestion in line 405-406, which recommended using five trials for each testing item for preschool children to remove the potential learning effect from the first few attempts. However, I think the author's statement is difficult to understand and accept in abstract(line 37 to 40), which is “Hence, NPFM guidelines should be revised by adding adequate familiarization sessions and standardized warm-up protocols as well as increasing the number of testing trials. SWC and MDC95 values of NPFM tests should be considered to realize true performance changes”. The main purpose of this manuscript was to investigate the reliability, sensitivity and minimum detectable change values of NPFM in preschool children, there was few result supported the author's point. So, it should be deleted in the abstract.

In conclusion (line 437 to 444)，these sentences also should be deleted which were similar to the discussion section.

7. PLOS authors have the option to publish the peer review history of their article (what does this mean?). If published, this will include your full peer review and any attached files.

Reviewer #1: No

---

## [Author Response · Author response to Decision Letter 1]

29 Oct 2020

We thank the reviewers and the academic editor for their time and valuable suggestions. In this revised version we have answered all the questions raised by the reviewers and the editor. We hope that this revised manuscript meets the standard for publication in PLOS ONE. Below please find our point-to-point responses to reviewers.

Reviewer: 1

I agree with Hraski and the author's suggestion in line 405-406, which recommended using five trials for each testing item for preschool children to remove the potential learning effect from the first few attempts. However, I think the author's statement is difficult to understand and accept in abstract(line 37 to 40), which is “Hence, NPFM guidelines should be revised by adding adequate familiarization sessions and standardized warm-up protocols as well as increasing the number of testing trials. SWC and MDC95 values of NPFM tests should be considered to realize true performance changes”. The main purpose of this manuscript was to investigate the reliability, sensitivity and minimum detectable change values of NPFM in preschool children, there was few result supported the author's point. So, it should be deleted in the abstract.

In conclusion (line 437 to 444)，these sentences also should be deleted which were similar to the discussion section.

Response from authors: Deleted the line 37 to 40 of abstract and also line 437 to 444 of conclusion accordingly

Academic Editor:

In methods part of abstract, please also mention, when the study was conducted.

Response from authors: Added the time of test back to abstract

Also, please state the Six items of NPFM in the methods.

Response from authors: Added 

In main text, please state what is the meaning of (ICC 3, 1) as the readers may not be familiar with it.

Response from authors: full form of ICC3,1 added as line 168 “intraclass correlation coefficient with two-way mixed-effects model and single measurement (ICC3,1)”

The heading under results in the result section (line 193-195) and another one in line 220-22 are too long. Please make it short and clear.

Response from authors: the heading is trimmed as requested

In discussion, please do not repeat the statistical results.

Response from authors: most statistical results originally included in the brackets were deleted per request. Only those essential in the main text for discussion purpose is retained.

The conclusion presented in the abstract and in the main texts should be aligned.

Response from authors: per the request of reviewer, we have deleted the last sentence to avoid redundant/duplicated sentences.

---

## [Editor Report · Decision Letter 2]

2 Nov 2020

Intraday reliability, sensitivity, and minimum detectable change of National Physical Fitness Measurement for Preschool Children in China

PONE-D-20-20616R2

Dear Dr. Ho,

We’re pleased to inform you that your manuscript has been judged scientifically suitable for publication and will be formally accepted for publication once it meets all outstanding technical requirements.

Kind regards,

Subas Neupane

Academic Editor

PLOS ONE
---

## [Editor Report · Acceptance letter]

10 Nov 2020

PONE-D-20-20616R2 

Intraday reliability, sensitivity, and minimum detectable change of National Physical Fitness Measurement for Preschool Children in China 

Dear Dr. Ho:

I'm pleased to inform you that your manuscript has been deemed suitable for publication in PLOS ONE. Congratulations! Your manuscript is now with our production department. 

Kind regards, 

on behalf of

Dr. Subas Neupane 

Guest Editor

PLOS ONE